# Anemia Is a Strong Predictor of Wasting, Disease Severity, and Progression, in Clinical Tuberculosis (TB)

**DOI:** 10.3390/nu14163318

**Published:** 2022-08-12

**Authors:** Senait Ashenafi, Amsalu Bekele, Getachew Aseffa, Wondwossen Amogne, Endale Kassa, Getachew Aderaye, Alemayehu Worku, Peter Bergman, Susanna Brighenti

**Affiliations:** 1Department of Pathology, School of Medicine, College of Health Sciences, Tikur Anbessa Specialized Hospital and Addis Ababa University, Addis Ababa 70710, Ethiopia; 2Center for Infectious Medicine (CIM), Department of Medicine Huddinge, ANA Futura, Karolinska Institutet, 141 52 Huddinge, Sweden; 3Department of Internal Medicine, School of Medicine, College of Health Sciences, Tikur Anbessa Specialized Hospital and Addis Ababa University, Addis Ababa 70710, Ethiopia; 4Department of Radiology, School of Medicine, College of Health Sciences, Tikur Anbessa Specialized Hospital and Addis Ababa University, Addis Ababa 70710, Ethiopia; 5Department of Preventive Medicine, School of Public Health, College of Health Sciences, Addis Ababa University, Addis Ababa 70710, Ethiopia; 6Clinical Microbiology, Department of Laboratory Medicine (Labmed), ANA Futura, Karolinska Institutet, 141 52 Huddinge, Sweden; 7Infectious Disease Clinic, Karolinska University Hospital, 141 86 Stockholm, Sweden

**Keywords:** tuberculosis, hemoglobin, anemia, clinical symptoms, body mass index, wasting, inflammation, disease severity, IL-6

## Abstract

A typical trait of chronic tuberculosis (TB) is substantial weight loss that concurs with a drop in blood hemoglobin (Hb) levels, causing anemia. In this observational study, we explored Hb levels in 345 pulmonary TB patients. They were divided into anemic or non-anemic groups which related to clinical symptoms, anthropometric measurements, and immune status. Data was obtained in a randomized controlled trial that we previously conducted using nutritional supplementation of TB patients in Ethiopia. A post hoc analysis demonstrated that anemic patients have a higher composite clinical TB score at baseline than non-anemic patients. Consequently, Hb values were significantly lower in underweight patients with moderate to severe disease and/or cavitary TB compared to normal weight patients with mild disease or non-cavitary TB. Anemia was associated with a low body mass index (BMI), low mid-upper arm circumference (MUAC), lower peripheral CD4 and CD8 T cells counts and IFN-γ levels, and a higher erythrocyte sedimentation rate (ESR). Chronic inflammation and TB disease progression appeared to be driven by elevated systemic levels of pro-inflammatory IL-6 in anemic patients. Multivariable modeling confirmed that a low Hb and a low BMI were key variables related to an unfavorable TB disease status. Although Hb levels increased with successful chemotherapy, anemic TB patients maintained a slower clinical recovery compared to non-anemic patients during the intensive phase treatment (two months). In conclusion, anemia is a strong predictor of wasting, disease severity, inflammation, and slower recovery in patients with pulmonary TB.

## 1. Introduction

Pulmonary tuberculosis (TB) remains one of the deadliest infectious diseases in the world, caused by the intracellular bacterium, *Mycobacterium tuberculosis* (Mtb). Diagnosis and follow up of TB disease is complex and is usually based on clinical symptoms as well as bacteriological confirmation and immunological tests. Active pulmonary TB is characterized by several typical clinical symptoms, such as a persistent cough, chest pain, fever, tachycardia, and weight loss. In addition, anemia is considered a risk factor for TB, and therefore anemia screening and diagnosis may contribute to improved anti-TB treatment and disease outcomes [1]. To facilitate the assessment of clinical symptoms in TB disease, a composite TB score has previously been generated and validated in different patient cohorts [2,3]. This is a numerical score composed of 11 variables, including conjunctiva pallor as a clinical indicator of anemia [4]. Similar to other chronic infections, TB is known to cause “anemia of inflammation,” which involves systemic inflammation and the release of cytokines, such as IL-6, IL-1, TNF-α, and IFN-γ, that may alter iron metabolism and reduce the number of red blood cells [1,5]. While these cytokines are required to activate immune cells and their effector functions to restrict TB infection, there are pathological side effects to these responses. Multiple mechanisms may be involved in anemia of TB disease, including loss of appetite resulting in poor nutrient uptake and impaired metabolism, or ineffective erythropoiesis [5,6]. Anemia of inflammation is diagnosed in patients with signs of systemic inflammation, such as an elevated erythrocyte sedimentation rate (ESR). However, the relation to iron deficiency is less clear, as these anemic conditions may co-exist in pulmonary TB patients due to increased blood loss from haemoptysis (blood in sputum) and/or malnutrition [5]. Accordingly, malnutrition and a low body mass index (BMI) has been shown to be associated with anemia [7] but also with more severe lung disease in pulmonary TB patients [8]. As such, malnutrition and low weight are risk factors for development of active TB but are also a consequence of progressive TB disease [9,10]. Malnutrition compromises immunity in different ways [11,12], which could result in decreased immune control and development of active TB disease. Therefore, wasting and decreased levels of hemoglobin in active TB patients may have direct effects on TB-associated morbidity and mortality, especially in developing countries [13].

In a randomized, controlled intervention trial conducted on pulmonary TB patients in Ethiopia, we used the clinical TB score as primary endpoint to evaluate the efficacy of nutritional supplementation of vitamin D_3_ (vitD_3_) and the short-chain fatty acid, phenylbutyrate (PBA) [14]. Here, in a post hoc analysis, we describe the association of anemia and blood Hb levels to the clinical TB score and select baseline variables including BMI, MUAC, vitD_3_, ESR, CD4, and CD8 T cell counts, as well as systemic levels of the T-cell-produced Th1 cytokine IFN-γ and pro-inflammatory IL-6.

## 2. Materials and Methods

### 2.1. Study Design

We performed a post-hoc analysis using an existing data set from a randomized, double-blinded, placebo-controlled trial that was conducted at the Chest Unit, Department of Internal Medicine, Tikur Anbessa Specialized University Hospital in Addis Ababa, Ethiopia, during the years 2013–2015 [14]. The original study was approved by ethical review boards in Ethiopia and Sweden, and the study was registered at www.clinicaltrials.gov (accessed on 2 October 2012), NCT01698476, prior to inclusion of the first patient. All patients and controls provided written and signed informed consent before enrolment and the study was conducted in accordance with the Declaration of Helsinki.

### 2.2. Patients

Pulmonary TB patients (*n* = 345) from the intention-to-treat cohort were in included in this post hoc study. Diagnoses were made from: a) positive sputum-smear microscopy or Mtb-culture, and/or b) clinical symptoms and chest X-ray findings consistent with TB, i.e., clinical TB defined according to WHO criteria. HIV-negative patients >18 years, with newly diagnosed pulmonary TB (<5 days chemotherapy) were included in the study. HIV-infection, multidrug-resistant TB (MDR-TB) or extrapulmonary TB, anti-TB treatment in the past 2 years, pregnant women, patients with kidney or liver disease, cancer, or autoimmune diseases were excluded from the study. In parallel with the clinical trial subjects, an untreated control cohort of *n* = 102 healthy individuals, was enrolled at the Tikur Anbessa Specialized University Hospital in Addis Ababa, Ethiopia, and used to assess baseline levels of secondary endpoint measures.

### 2.3. Interventions

The previous study was a two-arm intervention trial using daily adjunct therapy with vitD_3_ + PBA during the first 16 weeks of 6-months standard chemotherapy, including a fixed-dose combination of isoniazid, rifampicin, pyrazinamide, and ethambutol for 8 weeks (intensive-phase treatment) and isoniazid and rifampicin for an additional 16 weeks. Patients were randomized to receive daily oral 5000 IU vitD_3_ and 500 mg PBA or corresponding placebo tablets for 16 weeks. Vitamin D (Vigantoletten) and matching placebo tablets were donated by Merck Serono (Darmstadt, Germany), and PBA and matching placebo tablets were obtained from Scandinavian Formulas (Sellersville, PA, USA).

### 2.4. Clinical TB Score and Chest X-ray

To assess clinical symptoms and disease progression, we used a previously validated clinical TB score [2,3]. This is a numerical composite TB score (2-point scale: symptom absent (0 p) or present (1 p), max 13 p) which included self-reported clinical symptoms (cough, night sweats, and chest pain), as well as different variables monitored by the study physician upon clinical examination. These included conjunctival pallor, hemoptysis, dyspnea, tachycardia, positive findings at lung auscultation, fever, low BMI (<18 and/or <16), and low mid-upper arm circumference (MUAC) (<22 and/or <20 cm). Severity of TB disease was determined as mild (TB score 0–5) or moderate-severe (TB score >5) TB disease. Chest X-ray findings were used to determine cavitary TB or non-cavitary TB disease by visual examination of chest X-rays by an experienced radiologist.

### 2.5. Anthropometric and Laboratory Measurements

Patients were divided into anemic and non-anemic groups based on the normal reference blood Hb values for males (<13.5 g/dL) and females (<12 g/dL) [15]. Anthropometric measures included BMI (underweight ≤ 18.5 Kg/m^2^ or normal weight > 18.5 Kg/m^2^) and MUAC (underweight ≤ 21 cm or normal weight > 21 cm). Weight and height were measured to determine BMI (weight/(height^2^)) while a measuring tape was used to assess MUAC. Peripheral blood samples were collected for the described laboratory analyses at week 0, 4, 8, and 16. VitD_3_ (25(OH)D_3_) levels in plasma samples were determined at the Department of Clinical Chemistry, Karolinska University Hospital in Stockholm, Sweden, using a chemiluminescence immunoassay on a LIAISON-instrument (DiaSorin Inc., Stillwater, MN, USA), with a detectable range of 7.5–175 nmol L-1, CV 2–5%. Hemoglobin, white blood cell count (WBC) (Abbott, IL, USA), erythrocyte sedimentation rate (ESR), and CD4 and CD8 T cell counts (BD Biosciences, NJ, USA) were conducted at the International Clinical Laboratory (ICL) in Addis Ababa, Ethiopia, which is a Randox International Quality Assessment Scheme (RIQAS)-accredited and Centers for Disease Control and Prevention (CDC)-certified commercial laboratory. Mtb-specific IFN-γ production (IU) in whole blood samples were assessed using QuantiFERON-TB Gold in-Tube (Cellestis; Statens Serum Institut, Copenhagen, Denmark) analyzed at the Armauer Hansen Research Institute (AHRI) in Addis Ababa, Ethiopia, according to the manufacturer’s instructions. Levels of IL-6 (pg/mL) in plasma were quantified at the Center for Infectious Medicine (CIM), Karolinska Institutet, Sweden, using a Bio-Plex Pro Human Cytokine 27-plex Assay (Bio-Rad, Hercules, CA, USA). A multiplex analysis was performed on a sub-group of placebo-treated TB patients with mild (*n* = 10), moderate (*n* = 10), or severe (*n* = 10) TB disease and who were vitD_3_ deficient (25(OH)D_3_ levels in plasma < 50 nmol/L) at baseline.

### 2.6. Statistical Analysis

The sample size calculation was performed as previously described [14]. This post-hoc analysis included all patient samples from which blood Hb data at baseline and follow up (week 4, 8, and 16) was available (*n* = 345 and *n* = 294, respectively). The results are based on the intention-to-treat analysis, which included all subjects who fulfilled the inclusion/exclusion criteria and started the study treatment. Results were analyzed using GraphPad Prism 8, using an unpaired *t*-test or one-way ANOVA for normally distributed data, while data that did not pass a normality test were analyzed using the Mann–Whitney test. Parametric data is presented as mean or mean ± standard deviation (SD) or mean ±95% confidence interval (CI), while non-parametric data is presented as median ± interquartile range (IQR). For longitudinal analysis, a two-way Anova with Sidak’s multiple comparison test was used for normally distributed data, while a Kruskal –Wallis and Dunn’s post-test (for unpaired data) or Freidman’s test (for paired data) was used for analysis of data based on a small sample size. A *p*-value < 0.05 was considered significant. The independent effect of key baseline variables on the clinical TB clinical score was measured using a multivariable linear regression model, IBM SPSS Statistics 20.0, and Stata 13 (StataCorp, College Station, TX, USA).

## 3. Results

### 3.1. Baseline Characteristics

Baseline data from the study cohort is presented in Table 1. Both pulmonary TB patients and healthy controls had a slight over-representation of males (58–59%) compared to females (41–42%). Among the TB patients, all baseline variables, including the TB score, were similar in males and females, except for Hb and 25(OH)D_3_ levels which were significantly higher in males, while CD4 T cells counts were higher in females. However, the corresponding differences were observed in male and female controls, suggesting that these differences were not specific to TB disease. The composite TB score showed an average of 5.56 p, and accordingly half of the TB patients were grouped into mild disease (0–5 p) while the other half grouped into moderate-severe disease (6–13 p). Chest X-ray findings disclosed that most patients had cavitary TB (75%). Furthermore, anemia was common and found in approxmately 43% of the TB patients at baseline. Nutritional status was assessed using BMI, MUAC, and plasma 25(OH)D_3_ levels, and suggested an overall poor condition of the TB patients including significantly lower BMI and MUAC compared to the controls, while most of the TB patients (81%) as well as controls (89%) suffered from a vitD_3_ deficiency (25(OH)D_3_ < 50 nmol/L). Active TB disease was also associated with elevated ESR, WBC, and IFN-γ, but reduced CD4 and CD8 T cell counts in blood, suggestive of an ongoing systemic inflammatory response.

### 3.2. TB Disease Severity Is Associated with Blood Hb, BMI, and MUAC

The composite clinical TB score was used to determine disease severity among the patients and demonstrated a mean score of 3.72 in mild TB compared to 7.51 in moderate-severe TB (*p* < 0.0001) (Figure 1A). There was a significant (*p* < 0.0001) decrease in BMI (Figure 1B) and MUAC (data not shown) but also of blood Hb (Figure 1C) in moderate-severe TB, while ESR levels were significantly (*p* < 0.049) higher in moderate-severe compared to mild TB (Figure 1D). IFN-γ remained similar in these groups (Figure 1E). Accordingly, Hb levels were significantly lower (*p* = 0.032) in cavitary TB disease (Figure 1F) and Hb levels were also lower in underweight TB patients with a BMI ≤ 18.5 (*p* = 0.044) and a MUAC ≤ 21 (*p* < 0.0001) (Figure 1G,H). Using the dual combination of a low blood Hb (below the normal reference value for males and females, respectively) with a low BMI (underweight ≤ 18.5), it was revealed that less than 10% of mild TB patients, but more than 40% of patients with moderate-severe TB disease, expressed this combination (χ^2^ = 122, d.f. = 2, *p* < 0.0001) (Figure 1I). Contrarily, a normal Hb and BMI was evident in almost 60% of mild TB cases but only 5% of moderate-severe TB (Figure 1I), which suggests that the combination of low Hb + low BMI may comprise a simple but reliable measurement of TB disease severity. Importantly, other variables in the TB score, such as cough length or tachycardia, nor other parameters, such as smear-positivity, BCG vaccination, smoking, or respiratory rate, were not statistically different when comparing mild and moderate-severe TB disease (data not shown). Together, these results suggest that blood Hb and nutritional status have a fundamental role in determining the severity of TB disease.

Consistent with the results in Figure 1A–E, correlation analyses showed a significant (*p* < 0.0001) negative correlation between the clinical TB score and BMI (r = −0.64) as well as MUAC (r = −0.69, data not shown) and blood Hb (r = −0.41), but a positive correlation with ESR (r = 0.15, *p* < 0.0047) (Figure 2A–C). Contrary, blood Hb correlated to BMI (r = 0.14, *p* < 0.0098) and MUAC (r = 0.30, *p* < 0.0001, data not shown), but was inversely correlated to ESR (r = −0.31, *p* < 0.0001) (Figure 2D,E). As expected, BMI was strongly associated with MUAC (r = 0.76, *p* < 0001) (Figure 2F). Altogether these results demonstrate that TB disease severity is associated with low blood Hb values and a low BMI, as well as low MUAC, while elevated ESR reflected a high TB score and progression of disease.

### 3.3. Anemia Is a Strong Predictor of Moderate-Severe TB Disease

Our results suggested that blood Hb levels are an important determinant of disease severity in TB. To further characterize TB disease in patients based on Hb levels, patients were divided into non-anemic and anemic sub-groups using normal reference values for males and females (Figure 3A). Consistently, the TB score was significantly higher (*p* < 0.0001) in anemic patients (Figure 3B), whereas BMI (*p* = 0.0002) and MUAC (*p* < 0.0001) were significantly lower in anemic compared to non-anemic TB patients (Figure 3C,D). Anemic patients also showed skewed immune markers, including significantly elevated ESR (*p* = 0.0003) but reduced peripheral CD4 (*p* = 0.0053) and CD8 (*p* = 0.015) T cell counts. Furthermore, they showed decreased IFN-γ responses (*p* < 0.047) (Figure 3E–H), suggestive of a more severe and progressive TB disease in anemic patients.

Multivariable modeling was used to assess the relationship between the clinical TB score and key baseline variables, and demonstrated that blood Hb and BMI were statistically significant (*p* < 0.0001) predictors of TB disease severity (Table 2).

We then studied the longitudinal changes of the clinical TB score in anemic and non-anemic placebo-treated patients before and after four months of standard chemotherapy (Figure 4A). The TB score was significantly higher in anemic TB patients at baseline (*p* < 0.0001) but also at weeks four (*p* = 0.0028) and eight (*p* = 0.038) after start of treatment (Figure 4A). Blood Hb levels were steadily increasing with successful anti-TB therapy, particularly in anemic TB patients (Figure 4B). Nevertheless, blood Hb in anemic patients remained significantly lower (*p* < 0.0001) compared to non-anemic patients at each follow up time-point (Figure 4B). However, the number of anemic patients declined rapidly with treatment (*p* < 0.0001) from 43% at week zero to 6.8% at week 16 (Figure 4C). Assessment of Hb levels in TB patients based on their vitD_3_ status at baseline [16] showed a significant decline of Hb in patients with a severe vitD_3_ deficiency (25(OH)D_3_ < 30 nmol/L; *p* = 0.022 and 0.0098) (Figure 4D). Accordingly, blood Hb was associated with plasma 25(OH)D_3_ levels (r = 0.23, *p* < 00001, data not shown). However, there was no difference in 25(OH)D_3_ levels comparing non-anemic and anemic placebo-treated patients at baseline or follow up at weeks four, eight, and 16, although there was a relative increase in vitD_3_ levels in both groups after successful chemotherapy (Figure 4E). Accordingly, there was no difference in the TB score or number of anemic patients (Figure 4F) when comparing non-anemic and anemic TB patients who received either placebo or vitD_3_ + PBA treatment, which suggests that nutritional supplementation of vitD_3_ + PBA did not improve anemia in patients with pulmonary TB.

### 3.4. Plasma IL-6 Is Associated with Active TB Disease, Particularly in Anemic TB Patients

Plasma levels of the inflammatory cytokine IL-6 were measured in a sub-group (*n* = 30) of placebo-treated TB patients and demonstrated significantly higher (*p* < 0.0001) IL-6 levels in patients with active TB compared to healthy controls (Figure 5A). Furthermore, IL-6 levels were relatively higher in moderate-severe TB (Figure 5B), but were significantly higher (*p* = 0.0107) in anemic compared to non-anemic patients (Figure 5C). The Th1 cytokine IFN-γ was also significantly higher in active TB compared to healthy controls (Figure 5D) and relatively higher in moderate-severe TB (Figure 5E). In contrast to IL-6 levels, IFN-γ was lower in anemic compared to non-anemic patients (*p* < 0.058) (Figure 5F). Peripheral IL-6 levels decreased, while blood Hb and IFN-γ levels increased with effective chemotherapy in this sub-group of TB patients (Figure 5G–I). Accordingly, IL-6 levels correlated with the clinical TB score (r = 0.49, *p* < 0.0073) but were inversely correlated to blood Hb (r = −0.56, *p* < 0.0017) and IFN-γ (r = −0.48, *p* < 0.0144) (Figure 5F,G). These data indicate that pro-inflammatory IL-6, but not IFN-γ, is related to anemia of inflammation and disease progression in TB.

## 4. Discussion

This post hoc analysis intended to investigate how clinical TB symptoms and disease progression of pulmonary TB patients before and after starting chemotherapy were associated with blood Hb and anemia. The results suggest that low blood Hb levels and anemia were highly prevalent in TB patients with moderate-severe TB disease, including cavitary TB and underweight patients. Accordingly, low blood Hb and anemia were associated with a higher clinical TB score and ESR levels, but with reduced BMI and MUAC, along with low CD4 and CD8 T cell counts and lower IFN-γ levels. Longitudinal assessment of the clinical TB score revealed that TB disease severity remained higher in anemic compared to non-anemic TB patients during the eight-week intensive-phase treatment, although blood Hb levels increased while the number of anemic TB patients decreased rapidly with effective chemotherapy. There was no difference in the number of anemic TB patients comparing adjunct vitD_3_ + PBA treatment with the placebo, which suggests that this intervention did not restore TB-associated anemia. Finally, anemic TB patients, as well as patients with moderate-severe TB disease, displayed higher plasma levels of pro-inflammatory IL-6, which declined upon chemotherapy. In contrast, IFN-γ production was lower in anemic patients, suggesting that even if both cytokines are up-regulated in active TB, IL-6 was specifically related to anemia of inflammation and progression of TB disease.

It has previously been reported that clinical assessment of pallor to detect anemia is usually poor [4]. In line with this notion, we confirmed that conjunctiva pallor may not be an effective or sensitive variable in the TB score. While conjunctiva pallor was detected in 14.5% of TB patients at baseline [14], anemia based on blood Hb levels was detected in 43% of patients. Among the 11 clinical variables included in the composite TB score, low BMI and low MUAC were the most prominent parameters associated with disease severity and low blood Hb levels, which suggests that low weight and malnutrition are strong traits of more severe TB disease. As shown in other reports [17], we found a strong correlation between BMI and MUAC. Thus, dual assessment of blood Hb levels and BMI may represent a simple and effective combination to evaluate TB disease severity and disease progression at baseline and follow up. This is important as cases of more complex and severe TB disease constitutes a patient group with higher risks of complications and treatment failure.

While the proportion and cause of anemia in TB patients may vary substantially in different cohorts [18], our study suggests that anemia is diagnosed in approximately half of pulmonary TB patients at baseline and mostly involves mild to moderate anemia, with few cases of severe anemia. A recent report supports the concept that anemia of inflammation is the most prevalent cause of anemia in pulmonary TB patients, and consequently blood Hb levels were significantly raised after 30 to 60 days of anti-TB therapy [19]. The study showed that lower blood Hb levels were associated with higher numbers of acid-fast bacilli in sputum [19], which is consistent with our findings of a slower decline in clinical TB symptoms in anemic patients. Unfavorable treatment outcomes and a heightened degree of inflammatory perturbation has also been found in anemic patients with TB-HIV co-infection [20]. Thus, anemia per se seems to be a risk factor for more severe TB disease and for slower clinical (as well as bacteriological) recovery in response to chemotherapy. It is to be determined if correction of anemia would decrease the susceptibility to Mtb infection and/or reduce the likelihood of more severe TB disease. While treatment of the infectious agent causing chronic inflammation may be the most effective treatment of anemia, it is not known how improvement of anemia by nutritional supplementation could alter TB disease susceptibility and outcome. We have previously demonstrated that nutritional supplementation with vitD_3_ + PBA resulted in a significant reduction in the clinical TB score compared to the placebo, which correlated to reduced acid-fast bacilli in sputum at week four [14]. From our current results, it is apparent that adjunct treatment with vitD_3_ + PBA could not restore anemia, but elevated blood Hb levels and a concomitant reduction in anemic patients with chemotherapy was evident in both intervention and placebo groups. Thus, lower Hb levels in TB patients with a vitD_3_ deficiency (25(OH)D_3_ levels below 50 nmol/L) is likely a consequence of poor nutritional status, including low vitD_3_ levels in patients with more severe TB disease. As such, recent studies demonstrated that vitD_3_ deficiency was associated with malnutrition [21] and wasting [22]. Our findings suggest that nutritional supplementation with vitD_3_ + PBA could not improve TB disease severity by improving anemia, but ameliorates TB disease by other means, potentially by enhancing antimicrobial effector functions in Mtb-infected cells [23] or by modulating inflammation [24]. A systematic review from 2016, including 35 interventional trials based on macro- or micronutrient supplementation of 8283 TB patients, could not find consistent evidence that improved nutrition would support TB recovery [25]. Nevertheless, experiences from the pre-antibiotic era suggest that good nutrition could indeed improve pulmonary TB disease [26]. Accordingly, several studies have shown that there is a strong and consistent inverse relationship between BMI and development of TB disease [27,28,29], and data even suggests that obesity may be a protective factor for active TB [28]. Clearly, further investigations are needed to explore the biological mechanism linking low weight with immune failure and TB risk. Future research should also evaluate the prophylactic and therapeutic potential of nutritional interventions for a reduction in anemia and disease severity in pulmonary TB patients.

An increased understanding of the pathogenesis of anemia of inflammation could promote the development of targeted therapies aiming to reduce TB-associated anemia in the future. Potential pathophysiological mechanisms of anemia in chronic inflammation may involve decreased iron absorption, restricted iron availability, and inhibition of erythropoiesis [6]. In chronic infections such as TB, cytokines are key drivers of inflammation-associated anemia. This includes IL-6, which can stimulate hepatic expression of the acute-phase protein hepcidin, known to block iron uptake in the intestine and to reduce iron release by macrophages [6,30]. Accordingly, enhanced hepcidin levels result in an iron deficiency in the blood which inhibits erythropoiesis. Intriguingly, Mtb requires iron to survive and replicate within host macrophages [31]. Thus, hepcidin may contribute to iron deficiency in the blood and anemia as well as enhanced intracellular growth of Mtb, by virtue of increased levels of intracellular iron. In contrast to IL-6, IFN-γ has been shown to reduce hepcidin secretion and promote the export of iron out from Mtb-infected macrophages, which decrease iron availability and Mtb replication in infected cells [32]. In this study, we found that both IFN-γ and IL-6 were clearly elevated in active TB patients and in patients with moderate-severe disease, while only IL-6 was significantly higher in anemic TB patients. It has previously been demonstrated that high IL-6 concentrations and a high HIV viral load are associated with anemia in TB/HIV co-infected patients [33], which supports the notion that IL-6 aggravates anemia in pulmonary TB. Another report showed a strong correlation of serum IL-6 and hepcidin in patients with anemia of chronic disease, but not in patients with iron deficiency anemia [34]. Recently, an observational trial demonstrated that high IL-6 and hepcidin levels at baseline were related to low blood Hb and iron absorption in TB patients, which was reversed upon chemotherapy and resolution of TB-associated anemia of inflammation [35]. Previously, iron supplementation given to pulmonary TB patients with mild or moderate anemia initially enhanced the hematological status but failed to improve clinical or radiological symptoms [36]. Attempts to use an IL-6-blocking antibody to reduce hepcidin levels in lung cancer may improve cancer-related anemia [37,38] and anemia of inflammation in lymphoproliferative disorders [39]. Whether inhibition of IL-6-mediated inflammation in the early phases of pulmonary TB would be beneficial to correct anemia and support chemotherapy remains to be determined.

A strength of this study is the relatively large sample size and controlled assessment of primary and secondary outcomes in the context of a randomized controlled intervention trial. This enabled longitudinal quantification of clinical symptoms, and demonstrated how these were associated with the severity and progression of TB disease and with both nutritional as well as immunological status. A limitation of this study is that measures of iron-deficiency anemia were not performed including plasma ferritin, hepcidin, soluble transferrin receptor, and transferrin. In addition, it would have been informative to determine blood Hb levels and the proportion of anemia in the healthy control cohort. Currently, we are investigating markers of iron homeostasis together with a cytokine profile and chronic inflammation, as well as nutritional status, in patients with multidrug-resistant TB and controls in Ethiopia. This will contribute to an enhanced understanding of the pathological mechanisms of anemia in pulmonary TB.

## 5. Conclusions

In TB high-burden countries, screening and treatment of anemia and malnutrition may promote a more effective standard chemotherapy that could contribute to reduced transmission and TB related morbidity [40]. Our results suggest that low blood Hb levels in combination with low BMI provides a good measurement of TB disease state and prognosis. Chronic inflammation seems to be the primary cause of anemia in pulmonary TB patients and appears to be driven by elevated systemic levels of IL-6, but not IFN-γ. Adjunct interventions that reduce inflammation and/or malnutrition in active TB are likely most effective to restore anemia and to enhance disease recovery, especially in patients with severe TB disease.

## Figures and Tables

**Figure 1 nutrients-14-03318-f001:**
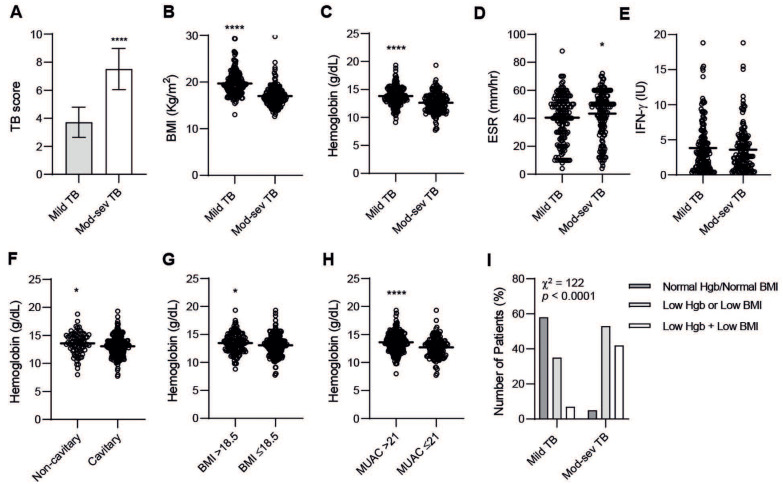
Baseline variables in TB patients with mild versus moderate-severe disease including (**A**) clinical TB score, (**B**) BMI, (**C**) blood Hb, (**D**) ESR, and (**E**) IFN-γ, and blood Hb levels in TB patients based on (**F**) chest X-ray findings, (**G**) BMI, or (**H**) MUAC. Combined blood Hb and BMI are shown in (**I**). Data (mean or mean ± SD) are presented in bar graphs or scatter plots and were analyzed using an unpaired *t*-test, *p* < 0.05 *, *p* < 0.0001 ****. Data in (**I**) were analyzed using a Chi-square test.

**Figure 2 nutrients-14-03318-f002:**
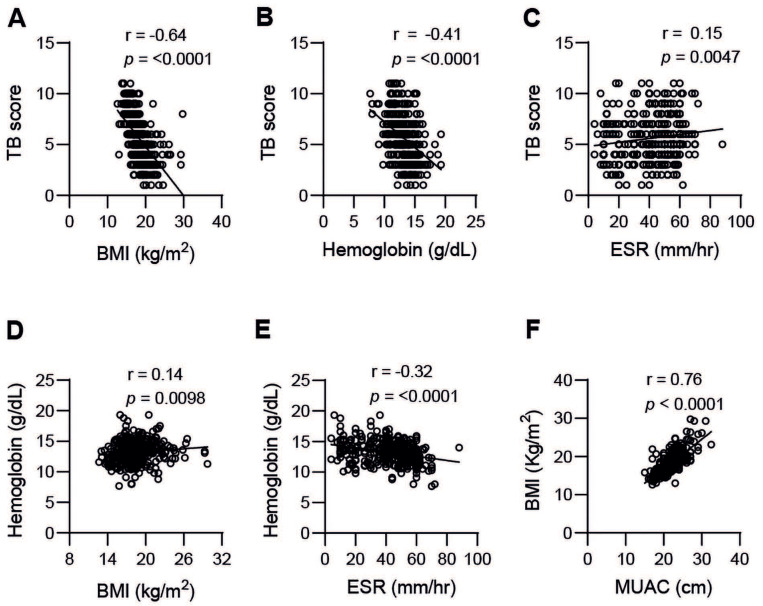
Correlation analyses between the clinical TB score and (**A**) BMI, (**B**) blood Hb, and (**C**) ESR, or blood Hb and (**D**) BMI, or (**E**) ESR. There was a strong correlation between BMI and MUAC (**F**). Correlation was determined using Spearman´s correlation test.

**Figure 3 nutrients-14-03318-f003:**
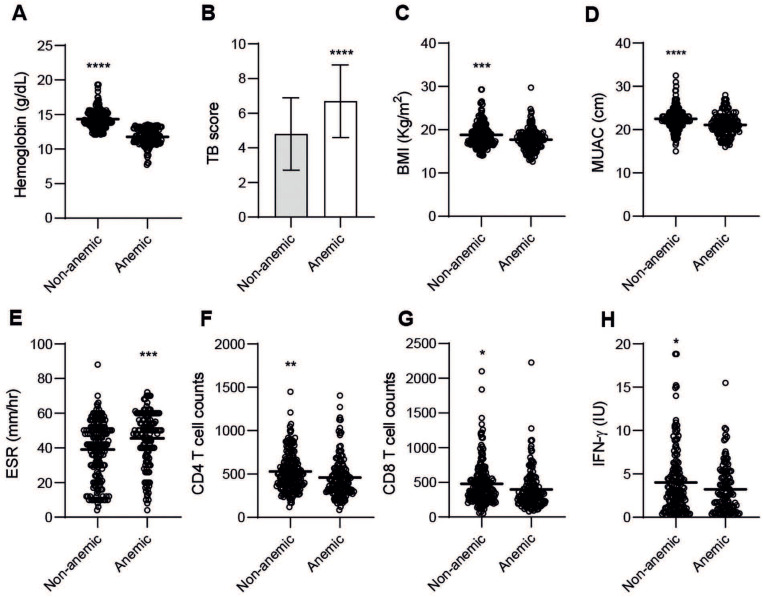
Baseline variables in non-anemic compared to anemic TB patients including (**A**) blood Hb, (**B**) clinical TB score, (**C**) BMI, (**D**) MUAC, (**E**) ESR, (**F**) CD4 T cell counts, (**G**) CD8 T cell counts, and (**H**) IFN-γ. Data (mean or mean ± SD) are presented in bar graphs or scatter plots and were analyzed using an unpaired *t*-test, *p* < 0.05 *, *p* < 0.005 **, *p* < 0.0005 ***, *p* < 0.0001 ****.

**Figure 4 nutrients-14-03318-f004:**
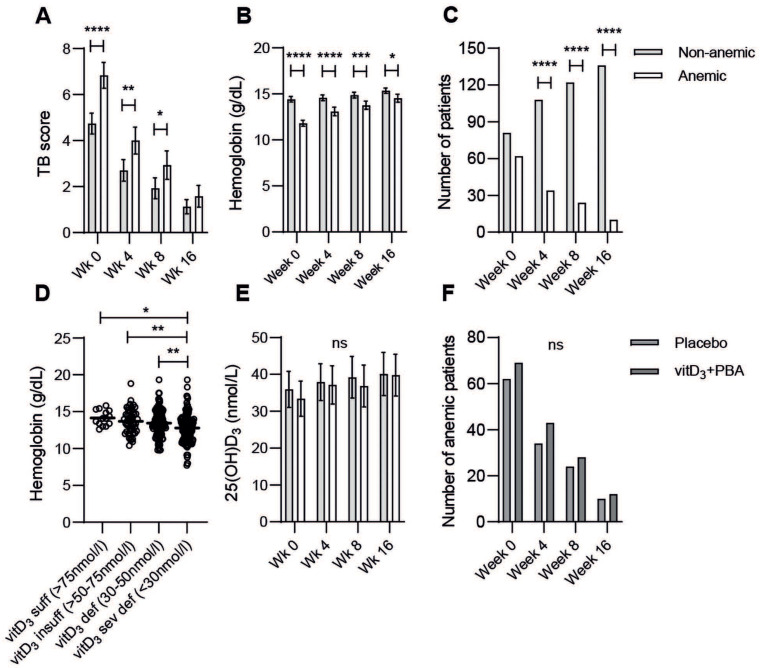
Longitudinal changes in non-anemic (light-grey bars) versus anemic (white bars) TB patients: (**A**) clinical TB score, (**B**) blood Hb levels, (**C**) the numbers of non-anemic and anemic TB patients, (**D**) blood Hb levels in TB patients based on their vitD_3_ status at baseline, (**E**) plasma vitD_3_ levels, and (**F**) the number of anemic TB patients in the placebo group (grey bars) or in the intervention group (dark-grey bars) at weeks zero (baseline), four, eight, and 16. Data (mean ± 95% CI) from placebo-treated TB patients (*n* = 143) are shown in bar graphs and were analyzed using a two-way ANOVA and Sidak´s multiple comparisons test, while Hb levels (mean) in TB patients (*n* = 345) with sufficient, insufficient, deficient, or severely deficient 25(OH)D_3_ levels are shown in the dot plot graph and were analyzed using a one-way ANOVA and Holm–Sidak´s multiple comparisons test, *p* < 0.05 *, *p* < 0.01 **, *p* = 0.0001 ***, *p* < 0.0001 ****, ns = not significant.

**Figure 5 nutrients-14-03318-f005:**
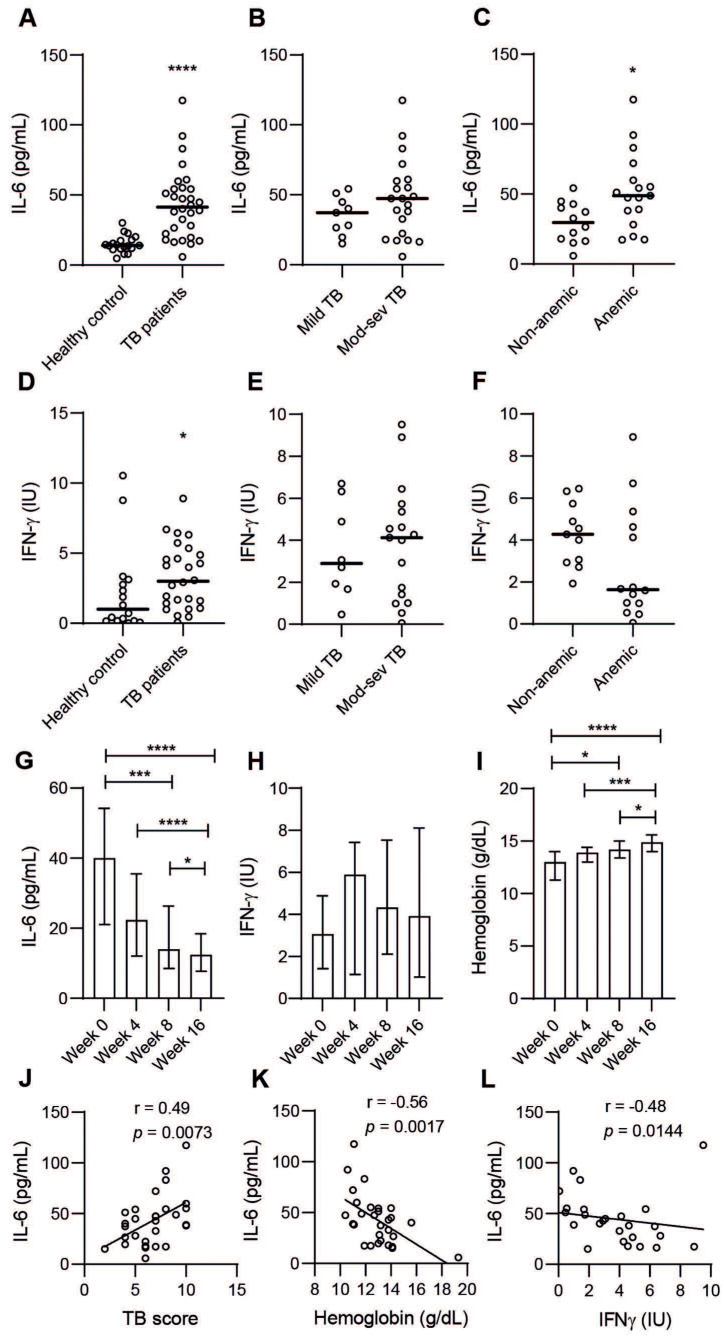
Peripheral IL-6 (**A**–**C**) and IFN-γ (**D**–**F**) levels in TB patients (*n* = 30) and healthy controls (*n* = 19) and in different sub-groups of TB patients. Longitudinal changes in (**G**) IL-6 levels, (**H**) IFN-γ levels, and (**I**) blood Hb levels in the TB patients at weeks zero (baseline), four, eight, and 16. Correlation analyses between IL-6 and (**J**) the clinical TB score, (**K**) blood Hb, and (**L**) IFN-γ levels as determined using Spearman´s correlation test. Data (median) from placebo-treated TB patients (*n* = 30) are shown in scatter plots and were analyzed using a Mann–Whitney test, *p* < 0.05 *, *p* < 0.0001 ****, while longitudinal data (median ± IQR) are shown in bar graphs and were analyzed using a Friedmans test, *p* < 0.05 *, *p* = 0.0001 ***, *p* < 0.0001 ****.

**Table 1 nutrients-14-03318-t001:** Baseline characteristics in pulmonary TB patients and healthy controls.

Variables ^a^	TB Patients (*n* = 345)	Healthy Controls (*n* = 102)	*p*-Value ^b^
Gender (M/F) (no/%)	200/145 (58/42)	60/42 (59/41)	0.880
Age (years)	30.4	36.2	<0.0001
TB score ^c^	5.61	na	
Mild TB/Mod-sev TB (no/%)	173/172 (50/50)	na	
Non-cavitary TB/Cavitary TB ^c^ (no/%) ^d^	82/238 (26/74)	na	
Hemoglobin (g/dL) (M/F) ^e^	13.22 (13.64/12.64)	nd	<0.0001
Non-anemic/anemic (no/%)	193/147 (57/43)	nd	
BMI (Kg/m^2^)	18.36	22.71	<0.0001
MUAC (mm)	21.87	25.83	<0.0001
25(OH)D_3_ nmol/L	35.28	32.47	0.190
ESR (mm/hour)	41.83	nd	
WBC counts (10^9^ cells/L)	7.87	nd	
CD4 T cell counts (cells/mm^3^)	499	668	< 0.0001
CD8 T cell counts (cells/mm^3^)	442	538	< 0.0032
IFN-γ (IU) ^f^	3.72	2.93	0.198

Abbreviations: mod-sev TB, moderate-severe TB; BMI, body mass index; MUAC, mid-upper arm circumference; 25(OH)D_3_, 25-hydroxyvitamin D; WBC, white blood cell count; ESR, erythrocyte sedimentation rate; na, not applicable; nd, not determined. ^a^ Data are *n* (%) or mean. ^b^ Unpaired *t*-test. Statistical significance between TB patients and healthy controls, except for blood Hb comparing the difference between males and females. ^c^ According to Wejse TB score classification (0–5 *p* = mild TB, >5 = moderate-severe TB) ^d^ Radiological findings on chest X-ray. Data were missing from *n* = 25 TB patients. ^e^ Normal reference value for males (<13.5 g/dL) and females (<12 g/dL). Blood Hb was not obtained from the healthy controls and data were also missing from *n* = 5 TB patients. ^f^ Determined using the QuantiFERON TB Gold In-Tube (QFTG) assay including *n* = 293 TB patients and *n* = 33 controls.

**Table 2 nutrients-14-03318-t002:** Independent effect of key variables on clinical TB score.

TB Score	*n*	Difference ^a^	95% CI ^b^	*p*-Value
Age	346	1.15	(−0.01 to 0.03)	0.253
Sex	346	−0.03	(−0.47 to 0.39)	0.868
Hemoglobin	341	−6.45	(−0.50 to −0.27)	<0.0001
BMI	346	−13.08	(−0.56 to −0.42)	<0.0001
ESR	336	1.75	(−0.00 to 0.02)	0.081
25(OH)D_3_	343	0.30	(−0.01 to 0.01)	0.761
CD4 T cell count	334	−0.85	(−0.00 to 0.00)	0.396
CD8 T cell count	332	0.74	(−0.00 to 0.00)	0.458

Abbreviations: CI, confidence interval; BMI, body mass index; ESR, erythrocyte sedimentation rate; 25(OH)D_3_, 25-hydroxyvitamin D. ^a^ Difference is quantified using the beta-coefficient. ^b^ Data are adjusted for gender, age, ESR, 25(OH)D_3_ levels, and CD4/CD8 T cell counts at baseline.

## Data Availability

All original clinical and laboratory data generated from this study has been documented in a Microsoft Access Database that is available with the corresponding author, Dr. Susanna Brighenti, at Karolinska Institutet.

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
