# Peer review of "Anemia Is a Strong Predictor of Wasting, Disease Severity, and Progression, in Clinical Tuberculosis (TB)"

_nutrients, 2022, doi:10.3390/nu14163318_

Round 1

Reviewer 1 Report

I read the manuscript, "Anemia is a strong predictor of wasting, disease severity and progression of clinical tuberculosis (TB)," with great interest. I believe this is a scientifically and well-written manuscript on the relation between anemia and TB disease manifestations and outcomes. I have a few minor comments:

1. Introduction, lines 79-81: there is no need to present the main result in the introduction

2. I am having my doubts about the methodology, I think it is better if the authors make it clear that there is no effect of vitamin D deficiency or interventions regarding vitamin D have a cofounding effect on the results of this study. I know it is unlikely, and I can see the clear results from the analysis, but I still find it useful to be clearly mentioned in the results/discussion.

3.  I believe SD would be a better fit than SEM for the presentation of the results

Author Response

Please, see the attachment below.

Reviewer 2 Report

From my point of view, the article is exciting. It respects the scientific rigor of presenting the results of well-conducted research. It also has a clinical impact by identifying risk factors for a severe form of tuberculosis. The importance of low Hb and a low BMI as unfavorable prognostic factors for the disease is emphasized. Prevention of risk factors can lead to a decrease in TB morbidity and mortality.

Author Response

Please, see the attachment below.
